# The Unfolded Protein Response Sensor PERK Mediates Stiffness-Dependent Adaptation in Glioblastoma Cells

**DOI:** 10.3390/ijms23126520

**Published:** 2022-06-10

**Authors:** Mohammad Khoonkari, Dong Liang, Marina Trombetta Lima, Tjitze van der Land, Yuanke Liang, Jianwu Sun, Amalia Dolga, Marleen Kamperman, Patrick van Rijn, Frank A. E. Kruyt

**Affiliations:** 1Department of Medical Oncology, University of Groningen, University Medical Center Groningen, Hanzeplein 1, 9713 GZ Groningen, The Netherlands; m.khoonkari@rug.nl (M.K.); d.liang01@umcg.nl (D.L.); ykliang12345@163.com (Y.L.); 2Polymer Science, Zernike Institute for Advanced Materials, University of Groningen, Nijenborgh 4, 9747 AG Groningen, The Netherlands; jianwu.sun@rug.nl (J.S.); marleen.kamperman@rug.nl (M.K.); 3Department of Molecular Pharmacology, Faculty of Science and Engineering, Groningen Research Institute of Pharmacy (GRIP), University of Groningen, 9713 AV Groningen, The Netherlands; m.trombetta.lima@rug.nl (M.T.L.); a.m.dolga@rug.nl (A.D.); 4Department of Biomedical Engineering-FB40, University of Groningen, University Medical Center Groningen, A. Deusinglaan 1, 9713 AV Groningen, The Netherlands; t.van.der.land.1@student.rug.nl; 5Department of Thyroid and Breast Surgery, Clinical Research Center, The First Affiliated Hospital of Shantou University Medical College, 57 Changping Road, Shantou 515041, China; 6W.J. Kolff Institute for Biomedical Engineering and Materials Science-FB41, University of Groningen, University Medical Center Groningen, A. Deusinglaan 1, 9713 AV Groningen, The Netherlands

**Keywords:** glioblastoma, extracellular matrix stiffening, tumor microenvironment, mechanical stress, PERK, unfolded protein response

## Abstract

Glioblastoma multiforme (GBM) is the most aggressive brain tumor in adults. In addition to genetic causes, the tumor microenvironment (TME), including stiffening of the extracellular matrix (ECM), is a main driver of GBM progression. Mechano-transduction and the unfolded protein response (UPR) are essential for tumor-cell adaptation to harsh TME conditions. Here, we studied the effect of a variable stiff ECM on the morphology and malignant properties of GBM stem cells (GSCs) and, moreover, examined the possible involvement of the UPR sensor PERK herein. For this, stiffness-tunable human blood plasma (HBP)/alginate hydrogels were generated to mimic ECM stiffening. GSCs showed stiffness-dependent adaptation characterized by elongated morphology, increased proliferation, and motility which was accompanied by F-Actin cytoskeletal remodeling. Interestingly, in PERK-deficient GSCs, stiffness adaptation was severely impaired, which was evidenced by low F-Actin levels, the absence of F-Actin remodeling, and decreased cell proliferation and migration. This impairment could be linked with Filamin-A (FLN-A) expression, a known interactor of PERK, which was strongly reduced in PERK-deficient GSCs. In conclusion, we identified a novel PERK/FLNA/F-Actin mechano-adaptive mechanism and found a new function for PERK in the cellular adaptation to ECM stiffening.

## 1. Introduction

Glioblastoma multiforme (GBM) is the most aggressive adult brain tumor with a high mortality rate and a patient survival rate of only around 6% 5 years after diagnosis [1]. Despite multimodal treatment with surgery, chemotherapy, and radiotherapy, the median overall survival rates remain less than 2 years. GBM tumors originate most often within the cerebral cortex, with a high tendency to invade to other parts of the brain, thus hampering surgical resection, which, together with therapy resistance, results in tumor recurrence and rapid progression [2,3].

Tumor heterogeneity is considered to be a main cause of therapy resistance in GBM, in which cancer stem cells (CSCs) are thought to play an important role [4,5]. CSCs, also identified in GBM, are highly malignant cells that drive tumor growth and progression, endowed with self-renewal, strong tumor initiation capacity, and high plasticity [6]. Cellular heterogeneity and plasticity are strongly regulated by the tumor microenvironment (TME). The TME is highly versatile and includes various stromal cell types, such as normal tissue cells, immune cells, endothelial cells, and different extracellular matrix (ECM) compositions with unique physiochemical properties. More recently, it has been recognized that in addition to biological cues, physical abnormalities also contribute to aggressive tumor behavior such as rapid proliferation, metastasis, and therapy resistance [7,8]. Stiffness (or rigidity) is an intrinsic tissue property that, together with solid stress, interstitial fluid pressure, and matrix architecture, has been categorized as one of four distinct physical traits of cancer [9]. The deposition and crosslinking of the ECM are the main causes of tissue stiffening and can occur locally, leading to heterogeneous stiffness.

The composition of the ECM in GBM alters during disease progression and is considered an important accelerator of malignancy [10,11,12]. The increased presence of hyaluronic acid (HA), brevican, glypican-1, neurocan, neuroglia protein 2 (NG2), proteoglycan, tenasican-C, and versican within the brain ECM during GBM progression highly affects the ECM physiochemical properties and gradually changes its structure, leading to ECM stiffening [13,14]. The normal brain tissue stiffness is in the range of 0.2 to 1.5 kPa, which gradually increases up to around 35 kPa in GBM [9,12,15,16,17]. Correlations between glioma grade and stiffness have been reported, with the ECM of GBM (grade 4) being stiffer than the ECM in lower-grade gliomas based on atomic force microscopy (AFM) analyses of tumor biopsies and magnetic resonance (MR) elastography in patients [18].

Hydrogels are commonly used as in vitro models to mimic ECM stiffening in tumors. Various types of hydrogels have been used to study the mechanical behavior of GBM cells. These include stiffness-tunable 2D or 3D models based on collagen, hyaluronic acid (HA), polyethylene glycol diacrylate (PEG-DA), gelatin methacrylate (GelMa), alginate, and chitosan [19,20,21,22]. Overall correlations have been found between increasing stiffness and the cellular adaptation of GBM cells, including cytoskeletal remodeling involving F-Actin, altered cell morphology, and cell migration and invasion [23,24,25,26,27,28]. The mechanisms by which mechanical cues translate into biochemical signals in tumor cells and affect their malignant behavior are increasingly studied, but not yet completely understood [29].

In the current study, we hypothesized that the unfolded protein response (UPR), particularly the protein kinase R (PKR)-like ER kinase (PERK) branch, is involved in the cellular adaptation of GBM stem cells (GSCs) to mechanical stress. The UPR is an adaptive quality control mechanism that maintains cellular protein homeostasis. Defects in proteins produced in the endoplasmic reticulum (ER) result in ER stress which is detected by three ER membrane localized sensors: inositol-requiring enzyme 1 α (IRE1), PERK, and activating transcription factor 6 (ATF6) [30]. Subsequently, these sensors trigger various adaptive molecular mechanisms in order to restore proteostasis and cell survival or activate cell death when ER stress is overwhelming [31]. In tumors, including GBM, different cell-intrinsic and extrinsic ER stressors have been identified which lead to UPR activation, including the production of misfolded proteins as a result of oncogenic transformation, nutrient deprivation, oxidative stress, and hypoxia [32]. Previously, we found that PERK plays a role in ER-stress-induced cytotoxicity in GSCs and showed that PERK regulates GSC differentiation by downregulating SOX2 via a yet-unknown noncanonical mechanism [32,33]. In addition, PERK-deficient GSCs displayed aberrant cell attachment and morphology upon differentiation induction when compared to wildtype counterparts [34]. Interestingly, alternative functions for PERK have been reported, such as in the secretory pathway and as a scaffold for binding to Filamin-A (FLNA), thereby controlling F-Actin remodeling [35,36]. Particularly, the reported function of PERK in F-Actin remodeling prompted us to study the potential functioning of PERK in the adaptive response to mechanical stress, as F-Actin remodeling is known to be a key player in this process [37,38].

For this, we generated and characterized a bioactive hydrogel made of human blood plasma (HBP)/alginate with tunable stiffness to mimic variable mechanical conditions in GBM. Patient-derived GSCs were used as a model to study GSC adaptation to increasing matrix stiffness and the role of PERK herein. Interestingly, PERK-deficient cells showed strongly impaired stiffness adaptation which included reduced FLNA expression, the absence of F-Actin remodeling, changes in cell morphology, reduced cell proliferation, and reduced motility. The PERK-FLNA axes, but not the classical UPR-PERK pathway, appeared to be instrumental in stiffness adaptation. Thus, for the first time, we report a novel function for PERK in regulating the adaptive response to increased ECM stiffness. These findings indicate that PERK has multiple functions in tumor biology, making it an interesting target for therapy.

## 2. Results

### 2.1. HBP/Alginate Hydrogel Characterization

HBP/alginate hydrogels with tunable stiffness were prepared as illustrated in Figure 1A,B. CaCl_2_ triggers the crosslinking of alginate and HBP via the fibrinogen/thrombin cascade, and tranexamic acid prevents the breakdown of fibrinogen crosslinks to improve gel stability. By varying the concentration of alginate from 0.0 to 1.81% *w*/*v*, different stiffnesses were obtained, resulting in hydrogels with stiffnesses ranging from 1.4 to 40 kPa (Figure 1C). This stiffness range reflected the physiological stiffness in the brain and GBM. The hydrogels were analyzed by determining water content, showing that water content decreased with increasing stiffness (Figure 1D). Scanning electron microscopy (SEM) was performed to analyze the structure of the hydrogels in relation to the alginate concentration and stiffness. As shown in Figure 1E, all gels displayed a fibrous structure with a slight increase in density upon increased stiffness. A denser structure at higher stiffness explains the observed reduced water-uptake ability when compared to softer gels.

### 2.2. GSCs Adapt to Increasing Matrix Stiffness by Changing Cell Morphology, Increased F-Actin Expression, and Cell Proliferation

Matrix stiffening is known to result in increased F-Actin expression in cells [27,39]. The cellular adaptation of GG16 cells to different matrix stiffness was investigated. Confocal microscopy analyses of GG16-LVGFP cells showed a change in cell morphology from a more round to an elongated phenotype upon increasing stiffness (Figure 2A). Increasing stiffness resulted in concomitant increases in F-Actin levels and remodeling that was also determined by F-Actin surface area quantification, showing a gradual increase in F-Actin polymerization with increasing stiffness (Figure 2B). The change in cell morphology was also quantified, with around 5% of cells being elongated in the softest matrix and up to 41% in the stiffest matrix (Figure 2B). F-Actin localization also showed drastic changes, with the monolithic aggregation of F-Actin around the cell plasma membrane upon increasing hydrogel stiffness (Figure 2C). Furthermore, a stiffer matrix increased GG16-LVGFP cell proliferation, which was around 30% higher at the highest stiffness compared to the lowest stiffness (Figure 2D). Together, these results indicate that GG16 cells adapt to increasing stiffness by strongly elevating F-Actin polymerization in the cytoplasm, also close to the cell membrane, which occurs concomitantly with elongated cell morphology. Cell proliferation also was stimulated by a stiffer matrix.

### 2.3. PERK Mediates Cellular Adaptation of GSCs to Increased Matrix Stiffness

Based on its known role in cellular ER stress adaptation and as a regulator of F-Actin remodeling [36], we examined the possible involvement of PERK in the adaptive response to matrix stiffness. For this, we compared the cellular adaptation of GG16-WT with PERK-KO counterparts using confocal immunofluorescent (IF) microscopy (Figure 3). Similar to for GG16-LVGFP cells, increasing stiffness resulted in elongated cell morphology and the concomitant accumulation of F-Actin in GG16-WT cells (Figure 3A,B,D; see also Appendix A). In addition, in another GSC model, GG6, increasing stiffness also correlated with increased F-Actin expression and cell elongation (Appendix A). Interestingly, cell elongation and F-Actin accumulation upon increasing stiffness was hardly observed in GG16-PERK-KO cells. Only at the highest stiffness (40 kPa) could F-Actin be detected (Figure 3E,F,H).

Previously, PERK, through interaction with Filamin-A (FLNA), was shown to regulate F-Actin remodeling to facilitate ER–plasma membrane interactions and restore calcium homeostasis [36]. Therefore, we hypothesized that the inability of PERK-deficient cells to adapt to increasing matrix stiffness is caused by aberrant FLNA functioning, leading to failed F-Actin remodeling. FLNA expression was examined in relation to matrix stiffness and PERK expression. In GG16-WT cells, FLNA showed cytoplasmic/plasma membrane localization that increased simultaneously with increasing stiffness. FLNA and F-Actin showed largely overlapping staining patterns, as predicted (Figure 3B,C). Similar observations for FLNA and F-Actin were made in GG6 cells (Appendix A). In addition, PERK expression also increased in a stiffness-dependent manner and overlapped partially with FLNA (Appendix A). However, in GG16-PERK-KO cells, the FLNA expression pattern was strikingly different and only showed detectable FLNA at the highest stiffness (Figure 3F,G; see also Appendix A). Together, these results indicate that PERK via FLNA regulates F-Actin remodeling in an increasing matrix stiffness.

### 2.4. Blocking F-Actin Polymerization in GG16 Cells Mimics the PERK-Deficient Phenotype by Failing to Adapt to Matrix Stiffness

To confirm that the observed inability of PERK-deficient cells to adapt to increasing matrix stiffness is caused by impaired F-Actin remodeling, we examined whether the direct inhibition of F-Actin polymerization resulted in a similar phenotype. GG16-WT cells were seeded on hydrogels with increasing stiffness and exposed to latrunculin B, an inhibiter of F-Actin polymerization. Latrunculin B potently inhibited F-Actin polymerization, as indicated by a sharp decrease in F-Actin expression compared to untreated controls (Figure 3A,B and Figure 4A,C). Indeed, latrunculin B prevented stiffness-dependent changes in cell morphology, whereas FLNA expression patterns were similar to those in untreated cells (Figure 3A,C,D and Figure 4A,B,D). Furthermore, in latrunculin-B-treated cells, PERK levels increased in a stiffness-dependent manner, similar to the trend seen in untreated controls (Figure 4E,F and Appendix A). Together, these findings show that interfering with F-Actin remodeling during stiffness adaptation inhibits cellular adaptation in a similar manner to that seen in PERK-deficient cells.

### 2.5. Exploring the Involvement of UPR PERK Signaling in Cellar Adaptation of GSC to Increasing Matrix Stiffness

Next, we examined the possible involvement of the UPR PERK signaling pathway in cellular adaptation to stiffness. GG16-WT cells were treated with GSK414 to inhibit PERK kinase activity. Figure 5A shows that GSK414 decreased both FLNA and F-Actin expression. When compared to untreated cells (Figure 3A), the FLNA levels were almost 5 times lower, and the F-Actin levels were around 4 times lower (Figure 5B,C). However, the expression levels were higher than in PERK-KO cells (Figure 3E) and stiffness-dependent increases in FLNA and F-Actin expression were still detected. In line with this, a change in cell morphology was also observed, although it was reduced compared to untreated cells, with visible cell elongation at the stiffest matrix (Figure 5A,D). Similar results were obtained in GSK414-treated GG6 cells (Appendix A). These results indicate that the inhibition of PERK kinase activity generates a similar but less aberrant phenotype compared to PERK-KO cells.

The downstream PERK effector, ATF4, was also examined for possible involvement in adaptation to stiffness. ATF4-deficient GG16 cells, GG16-ATF4-KO, were cultured on hydrogels with increasing stiffness. Figure 6 shows that the level of FLNA and F-Actin expression was in the same range as that detected in GG16-WT cells (Figure 6B,C,F,G). Additionally, stiffness-induced cell elongation was similar in the absence or presence of ATF4. (Figure 6D,H). These findings indicate that classical downstream UPR PERK-ATF4 signaling is not required for F-Actin polymerization and cellular adaptation to a stiffer matrix.

### 2.6. PERK Mediates Stiffness-Dependent GBM Cell Migration and Proliferation

Since F-Actin is involved in cytoskeletal remodeling, known to be associated with cell migration [40,41], we investigated whether PERK has an effect on cell motility. For this, GG16-WT and GG16-PERK-KO spheroids with similar sizes were seeded on hydrogels with three different matrix stiffnesses, 1.4, 28, and 40 kPa, and migration was monitored during 48 h. A clear invasive front was seen in GG16-WT spheroids, resulting in an expanding cell surface that was quantified after 48 h (Figure 7A,B). Notably, cell migration progressively increased significantly with hydrogel stiffness. Interestingly, PERK-deficient cells showed strongly reduced (around 2.6-fold) cell migration under all three stiffness conditions compared to WT cells, and a stiffness-dependent increase in migration was not detected (Figure 7C). These findings indicate that a deficiency in PERK resulted in reduced cell motility as well as the loss of the stiffness-dependent enhancement of cell migration.

Finally, the cell proliferation rates of GG16-WT cells increased with increasing matrix stiffness. After 6 days, the cell numbers on the softest matrix (1.4 kPa) were almost reduced by 50% when compared to the stiffest matrix (40 kPa). However, in the absence of PERK, cell proliferation was reduced overall, particularly with the highest stiffness (Figure 7D).

## 3. Discussion

In this study, we identified a novel function of PERK as a regulator of the cellular adaptation of GBM stem cells to matrix stiffness. A stiffness-tunable HBP/alginate gel was developed in the range of 1.4 to 40 kPa, reflecting the stiffness span reported in normal brains and GBM [15,16,42,43]. HBP/alginate hydrogels are not often used despite the fact that they have several favorable properties [44,45]. HBP is a natural complex mixture of many proteins and growth factors, is highly compatible with different polymers, and guarantees biocompatibility for use as an adherent scaffold for cell culture. Yet, a limitation of the gels is the inability to retrieve cultured cells to perform a number of techniques such as Western blotting or flowcytometry.

Our finding that increasing matrix stiffness stimulated GBM cell migration and proliferation is in agreement with previous studies. For example, Beliveau et al. used hydrogels with different collagen concentrations and showed that GBM cell migration highly depends on the tissue mechanics and structure and that aligned structures/confined spaces promote GBM cell invasion [21]. Erickson et al. showed that increasing stiffness with HA-based hydrogels from 1.4 to 28 kPa increased U87 GBM cell proliferation and temozolomide resistance [20]. The expression of several genes increased with stiffness, such as the HA receptor CD44, protease MMP-2, and the epithelial to mesenchymal transition regulator TWIST-1, known to facilitate cell migration and invasion. In this context, CD44 has been reported to mediate the HA-dependent cell adhesion of GBM cells and cell motility [46,47].

Here, we found that F-Actin expression and remodeling is strongly altered by the exposure of GSCs to increasingly stiff matrices, which is essential for cellular adaptation, since the blocking of F-Actin polymerization with latrunculin B inhibited this process. F-Actin is a known key player in mechano-transduction [37,38]. In fact, most of the mechano-sensing pathways first alter the cytoskeleton organization, followed by the activation of inter- and intracellular signaling pathways, where F-Actin functions as a hub [48]. F-Actin can bind to integrins and initiate cell–ECM interactions by forming intracellular connections [49]. The changes in F-Actin filament assembly affect F-Actin expression levels and directly have an effect on cell motility [50]. It is known that upon cell elongation, the cytoskeleton expands through the formation of actin lamellipodia and filopodia, thus facilitating cell motility, proliferation, and invasion [51]. We observed increased stiffness-dependent F-Actin expression and remodeling which was also localized around the cell membrane region, which is expected to facilitate cell movement. In agreement with this, we found a positive correlation between stiffness and increased GSC motility.

PERK has been reported to mediate F-Actin remodeling through interactions with FLNA in mouse embryonic fibroblasts in order to coordinate the formation of ER–plasma membrane contact sites to maintain calcium homeostasis [36]. Here, we provided evidence that F-Actin is likely remodeled via a similar PERK–FLNA mechanism, which is required for the cellular adaptation of GSC to increasing stiffness. FLNA is known to regulate F-Actin network organization and adaptation to mechanical stress [52,53]. In the absence of PERK, FLNA and F-Actin both showed a very low expression, with only some expression at the highest matrix stiffness, unveiling a role for PERK in mechanical stress adaptation and potentially as a regulator of cell–ECM interactions. FLNA and F-Actin expression showed strong colocalization in PERK-proficient GSCs, and PERK expression partially overlapped with FLNA. The stiffness-dependent changes in cell morphology, from round to elongated, are directly linked with cell motility. Elongated cells have higher motility and are reported to be more invasive, whereas rounded cells are less motile [27,51,54]. In the absence of PERK, there was almost no cell elongation detected from the softest to the stiffest hydrogels. This is in line with a strong reduction in F-Actin expression in PERK-deficient GSCs and decreased cell migration. An increasing stiffness and concomitant increase in F-Actin expression was also associated with enhanced cell proliferation in PERK-proficient GSCs when compared to PERK-deficient cells. Of note, another UPR sensor, IRE1, also was demonstrated to regulate cytoskeleton remodeling via FLNA interaction, in a mechanism independent from PERK [55]. Evidence was provided that IRE1 supports the phosphorylation of FLNA involving PKCalpha and stimulated cell migration. Whether IRE1 is involved in stiffness-dependent cellular adaptation remains to be investigated.

The classical UPR PERK-ATF4 axes were not required for the stiffness adaptation of GSCs, since ATF4-KO cells showed normal adaptive changes in FLNA/ F-Actin expression and cell morphology. However, PERK kinase activity was to some extent required for stiffness adaptation, because pharmacological PERK kinase inhibition partially decreased the ability of GSCs to adapt to matrix stiffening, as indicated by the detection of F-Actin remodeling and cell elongation, particularly at higher matrix stiffness compared to untreated controls. This finding suggests that PERK oligomerization and autophosphorylation facilitates FLNA-dependent F-Actin expression. Van Vliet et al. demonstrated that PERK-FLNA interactions induced by ER calcium depletion primarily required a scaffold function of PERK and did not require classical ER-stress-dependent PERK phosphorylation [36]. However, PERK dimerization and autophosphorylation was found to occur independently from ER stress and may facilitate binding to FLNA. Thus, we found a PERK/FLNA/F-Actin mechano-mechanism that is involved in stiffness adaptation; however, the precise molecular mechanism by which PERK via FLNA regulates F-Actin polymerization and its possible role in mechano-sensing remains to be further elucidated. A model summarizing our findings is shown in Figure 8.

The therapeutic relevance and implications of our findings remain to be further investigated. However, it is clear that the therapeutic modulation of PERK not only targets the UPR, but will also affect cancer cell stemness and plasticity, which also includes the adaptation to mechanical stress.

## 4. Materials and Methods

### 4.1. Preparation of Stiffness-Tunable Human Blood Plasma (HBP)/Alginate Hydrogel

Human blood plasma (HBP) was generated from 500 mL of human blood obtained from 5 healthy (voluntary) donors (approved by the ethics committee of the UMCG) by mixing it with an EDTA anticoagulant agent (BD, 366643, New Jersey, NJ, USA) for 15 min and centrifuging it at 2700 rpm for 15 min at 27 °C. Then, 300 mL of blood plasma was collected and centrifuged again in a similar way to deplete the cellular content. The HBP was diluted 1:1 with cell culture medium containing FBS 10% (Thermo Fischer Scientific, Hamburg, Germany), aliquoted, and stored at −20 °C. Alginate–sodium salt (Mw 250 Da) (Sigma-Aldrich, Darmstadt, Germany) was dissolved in DI-water at six different concentrations of 0.28, 0.56, 0.84, 1.12, 1.41, and 1.81% *w*/*v* by mixing the solutions for 24 h at room temperature. Solutions of CaCl_2_ (3.5% *w*/*v*), NaCl (0.9% *w*/*v*) and tranexamic acid (5 mg/mL) (all from Sigma-Aldrich, Darmstadt, Germany) were prepared in DI-water. All solutions were sterilized via autoclaving and stored at 4 °C for further use. Hydrogels were prepared by mixing a CaCl_2_ (3.5% *w*/*v*), NaCl (0.9% *w*/*v*), and tranexamic acid solution (5 mg/mL) with an alginate/HBP solution to yield hydrogels consisting of 50% HBP, 23% alginate solution, 14% NaCl, 12% CaCl_2_, and 1% tranexamic acid. After vortexing for 20 s, the mixtures were quickly transferred to the cell culture setup and incubated at 37 °C and 5% CO_2_ for 40 min to facilitate the gelation process. Hydrogels were generated in in-house-made polydimethylsiloxane (PDMS) molds. Briefly, a PDMS layer of 1 mm thickness was fabricated using Sylgard-188 (Sigma-Aldrich, Darmstadt, Germany) and punched to make eight circular holes with 6 mm diameter and 1 mm depth; this layer was glued (using unreacted silicone elastomer, followed by 30 min curing at 80 °C) on top of a glass slide and sterilized via autoclaving.

### 4.2. Stiffness Measurements

Hydrogel stiffness was determined using stress–strain analysis. For this, a Low Load Compression Tester (LLCT) device (manufactured at the biomedical engineering department, University Medical Center Groningen-UMCG, Groningen, NL) was used. A plunger (⌀ 2.5 mm) was set to compress the substance with a speed of 5 µm/sec and 1% strain. Hydrogels generated on a microscope slide within a surface area formed by a liquid blocker (liquid blocker pen, hydrophobic ink) were transferred to a Petri dish with DMEM medium and placed at 37 °C for 30 min prior to stress–strain analysis. Duplicate measurements were performed.

### 4.3. Rheology

Stiffness measurements obtained using LLCT were corroborated with an MCR 300 rheometer (Anton Paar Co., Oosterhout, The Netherlands) in parallel-plate mode, a fixed strain at 1%, and an oscillatory torque over a frequency range of 0.1–100 Hz.

### 4.4. Scanning Electron Microscopy (SEM)

Hydrogels were directly frozen in liquid-N_2_ for 20 min and freeze-dried at −40 °C under vacuum for 72 h. All steps were conducted immediately to reduce possible structural changes. The freeze-dried gels were carefully broken into pieces, and samples were gold-coated to perform scanning electron microscopy (SEM) (Nova nanoSEM 650).

### 4.5. Water Content of the HBP/Alginate Hydrogel

The dry and wet weight of each hydrogel was determined. One set of hydrogels (each being 100 µL in volume) was incubated with 100 µL of 1x-PBS for 48 h at 37 °C, and the other set was incubated for 48 h at 37 °C without PBS or any medium and yielded after weighing the wet or dry weight, respectively. The water content of the hydrogels was calculated using the following formula [56]:$$\mathrm{WC}\%=\left[\frac{\mathrm{Wwet}-\mathrm{Wdry}}{\mathrm{Wwet}}\right]*100$$

### 4.6. Cell Culture

GG16 and GG6 cells were previously isolated from GBM surgical samples [57]. Genetically modified variants of GG16-WT control (ctr) and GG16-PERK-KO and GG16-ATF4-KO were described previously [33]. GG16-LVGFP cells were generated via transduction with viral particles containing the pRRL-SFFV-IRES-EGFP lentiviral vector ([58], kindly provided by Dr. ATJ Wierenga, Dpt. of Hematology, University Medical Center Groningen, Groningen, The Netherlands). Cells were cultured under neurosphere/CSC conditions in neurobasal medium (NBM) with 2% B27 supplement, 20 ng/mL bFGF, 20 ng/mL EGF, and 1% L-glutamine (all purchased from Sigma-Aldrich, Darmstadt, Germany), named NBM^+^ medium. When indicated, the cells were treated with the chemical inhibitor GSK2606414 (GSK414) (TOCRIS, 5107, Bristol, UK) at 1 or 5 µM latrunculin B (Sigma Aldrich, Darmstadt, Germany) in culture medium. Cells were maintained in an incubator under a humidified atmosphere with 5% CO_2_ at 37 °C. For 2D cell culture on hydrogels, neurospheres were dissociated with accutase (Merk, Darmstadt, Germany), and single cells were counted. Then, 5000 cells in 50 µL of NBM^+^ were seeded as a droplet on top of each hydrogel. After allowing cell adherence for 4 h in an incubator at 37 °C, 1 mL of NBM^+^ was added to each well, and cell culturing was prolonged for seven days for further analyses. Cells were regularly tested for mycoplasma and authenticated using STR profiling.

### 4.7. Migration and Proliferation Assays

GBM spheroids were made with the hanging-droplet method in order to obtain spheroids with similar sizes. Briefly, 25 µL droplets with 2000 of the indicated cells in NBM^+^ with 2% bovine serum albumin (BSA) (Thermo Fischer Scientific, Waltham, MA, USA) were pipetted on a Petri dish, inverted, and placed in an incubator at 37 °C for 5 days. Then, the droplets were inspected using normal bright field microscopy (EVOS XL core imaging system), and spheroids were selected and carefully seeded on top of hydrogels with variable stiffnesses, generated in µ-Slide Angiogenesis cell culture chips (Ibidi GmbH, 81506, Munich, Germany), and placed in an incubator for 3 h to allow spheroid adherence. Next, 50 µL of NBM^+^ was added to each well, and cell migration was monitored using bright field microscopy imaging at 12 h intervals for a period of 48 h. Cells were fixed and permeabilized with 4% PFA and 0.5% Triton-x100, respectively, followed by the addition of mounting medium solution with DAPI^TM^ (Ibidi GmbH, 50011, Munich, Germany) and fluorescent imaging (EVOS XL core imaging system). To determine cell migration, all images were processed using ImageJ to determine cell migration by comparing the initial spheroid size with the migrated cell area. Spheroid edges at day 0 and day 2 (48 h) were marked with a particle parameter set at 0.2 to infinity, and spheroid areas were measured based on pixel per area.

A µ-Slide Angiogenesis cell culture chip (Ibidi GmbH, 81506, Munich, Germany) was used for the proliferation assay. Here, 1000 cells were labeled with Cell Tracker™ Green CMFDA Dye (Thermo Fischer scientific, c2925) and seeded on top of each gel with different stiffnesses. Cell proliferation was determined by analyses of the microscopic images obtained using bright field microscopy (EVOS XL core imaging system) during a time at 24 h intervals for up to 6 days. Cell numbers were determined with ImageJ software (cell counting feature). The cell numbers were normalized to cell numbers on day 1 and plotted.

### 4.8. Immunofluorescent Microscopy

For immunofluorescent (IF) confocal microscopy, cells on hydrogels were subsequently rinsed 3 times with PBS for 5 min, fixed with 4% paraformaldehyde (PFA) solution in PBS for 20 min, washed 3 times with PBS, and permeabilized with 0.5% Triton-x100 in PBS for 10 min. After being washed 3 times with PBS, cells were incubated with Alexa-fluor^TM^-Phalloidin (Alexa-594, Thermo Fischer-scientific, Waltham, MA, USA) at a 1:40 dilution in PBS for 60 min at room temperature to stain F-Actin. Alternatively, after permeabilization, cells were subsequently blocked with 3% BSA in 1x-PBS for 1 h, incubated with Filamin-A Monoclonal Antibody (Thermo Fisher Scientific, MA5-11705, Waltham, MA, USA) (1:100 in PBS) overnight at 4 °C, washed with PBS 3 times (5 min), incubated with Goat-anti-Mouse secondary antibody Alexa 647 (Thermo Fischer Scientific, A31571, Waltham, MA, USA) for 30 min in 1% BSA/0.1% Triton in PBS, washed 3 times with PBS, and mounted with the anti-fade Mounting liquid with DAPI^TM^ (Thermo Fischer-Scientific, Waltham, MA, USA) for microscopic analyses. Cells were stained for PERK, using primary antibody PERK (Cell Signaling, 5683, Amsterdam, The Netherlands) and secondary antibody Alexa 488 (Thermo-Fischer Scientific, A11008, Waltham, MA, USA). After being washed three times with PBS, cells were mounted with mounting Medium with DAPI^TM^ (Ibidi GmbH, 50011, Munich, Germany). Microscopic samples were kept at 4 °C until further analyses using the Leica SP8x (Leica Microsystems Co., Wetzlar, Germany) laser scanning confocal microscope. Images were obtained using the 63x oil immersion lens, and the intensity of the laser, gain, and saturation were kept the same for all the samples to generate comparable data. A scanning depth of 10 µm was used during microscopy to image both cells on top of the gels as well as cells penetrating the gels to ensure accurate imaging.

### 4.9. Cellular Characterization and Microscopic Data Analysis

Cell morphology was evaluated by eye, counting the number of rounded and elongated cells in each microscopy image. In addition, cell shape was also analyzed with LASX software (Leica Microsystem Co., Wetzlar, Germany). Briefly, for each individual cell, the length (highest measured value) was divided by the width (lowest measured value), in which measured ratios more than 1.5 (≥1.5) represented elongated cells, and measured rations less than 1.5 (1.5 ≤ X ≥ 1) represented rounded cells. The ratios of rounded and elongated cells are indicated in percentages.

The F-Actin surface areas obtained using confocal microscopy imaging were quantified with LAS-X software (Leica Microscopy Co., Wetzlar, Germany) and ImageJ software. Each image was analyzed separately while keeping all process conditions the same. Briefly, original images were switched to 8 bit images. The dimension was corrected in the scale section (set scale). The image was improved using the ImageJ plugins facility to decrease the background noise and blurriness of the image. The contrast and sharpness of the image were enhanced while keeping the color intensity/brightness the same. Using the image threshold, the F-Actin surface area was marked and measured in µm^2^, and averages were used for plotting. The F-Actin surface area was normalized to the average number of cells within the image. The same method was used to measure the surface area of Filamin-A and PERK. To measure the F-Actin surface area around the cell membrane, 5 single cells were picked in each image and analyzed using a newly developed phyton-based script added to the open-source ImageJ software. Depending on the cell size, specific areas of the cell membrane at a 20–30 µm distance from the center of the cell were selected to measure F-Actin. Averages from 5 measurements per cell were used to plot cell-membrane-localized F-Actin.

### 4.10. Statistical Analyses

Experiments were repeated at least three times unless otherwise indicated. OriginLab (2020b) software was used to plot the data and analyzed with the one-way ANOVA data analyses tool. Data are presented as means with standard deviations (SDs). A significant difference in statistics was considered at *p* < 0.05.

## 5. Conclusions

In conclusion, we identified a novel function for PERK as a regulator of the cellular adaptation of GBM cells to increasing matrix stiffness and the associated enhancement of proliferation and migration potential. The PERK/FLNA/F-Actin axes are required for this mechano-adaptive mechanism.

## Figures and Tables

**Figure 1 ijms-23-06520-f001:**
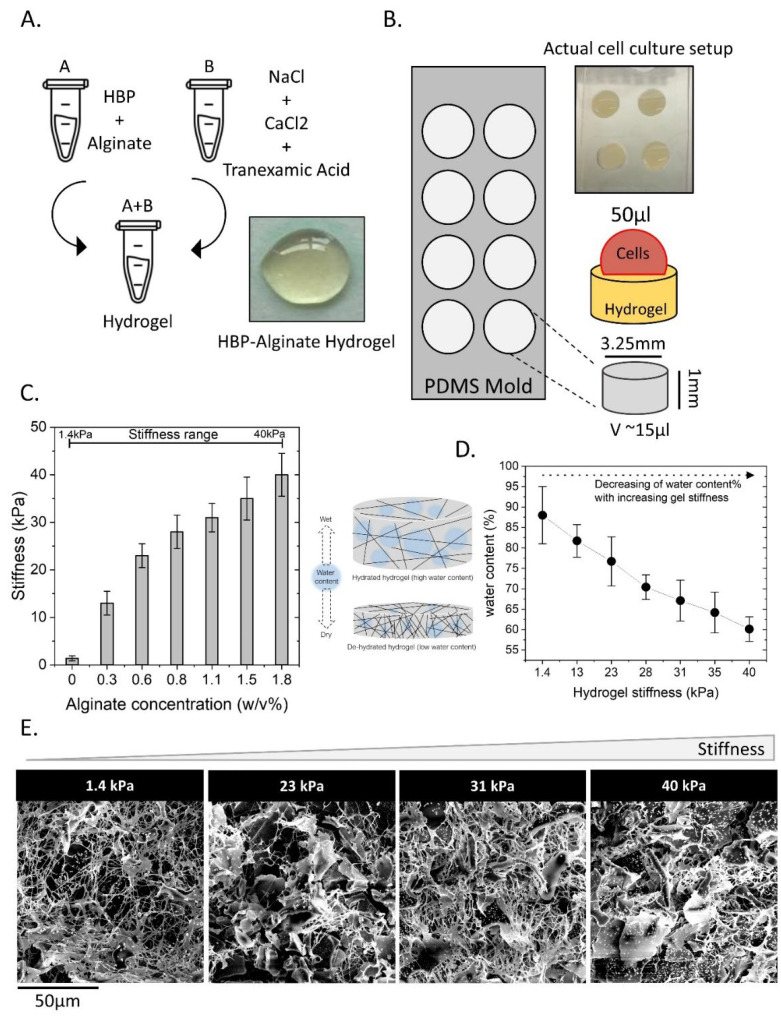
Preparation and characterization of human blood plasma (HBP)/alginate hydrogels with tunable stiffness. (**A**) CaCl_2_ was used as an ionic crosslinker for both alginate and HBP, and NaCl was added to minimize the free radical effect on gel degradation. Tranexamic acid was added to inhibit HBP decomposition. (**B**) Fabricated PDMS cell culture mold was used for hydrogel preparations. (**C**) By varying the alginate concentration, the stiffness was tuned in a range from soft (1.4 kPa) to stiff (40 kPa). (**D**) The water content (%) of the hydrogels was determined in relation to stiffness. (**E**) Hydrogel structure was determined using scanning electron microscopy (SEM) images from the hydrogels with different stiffnesses.

**Figure 2 ijms-23-06520-f002:**
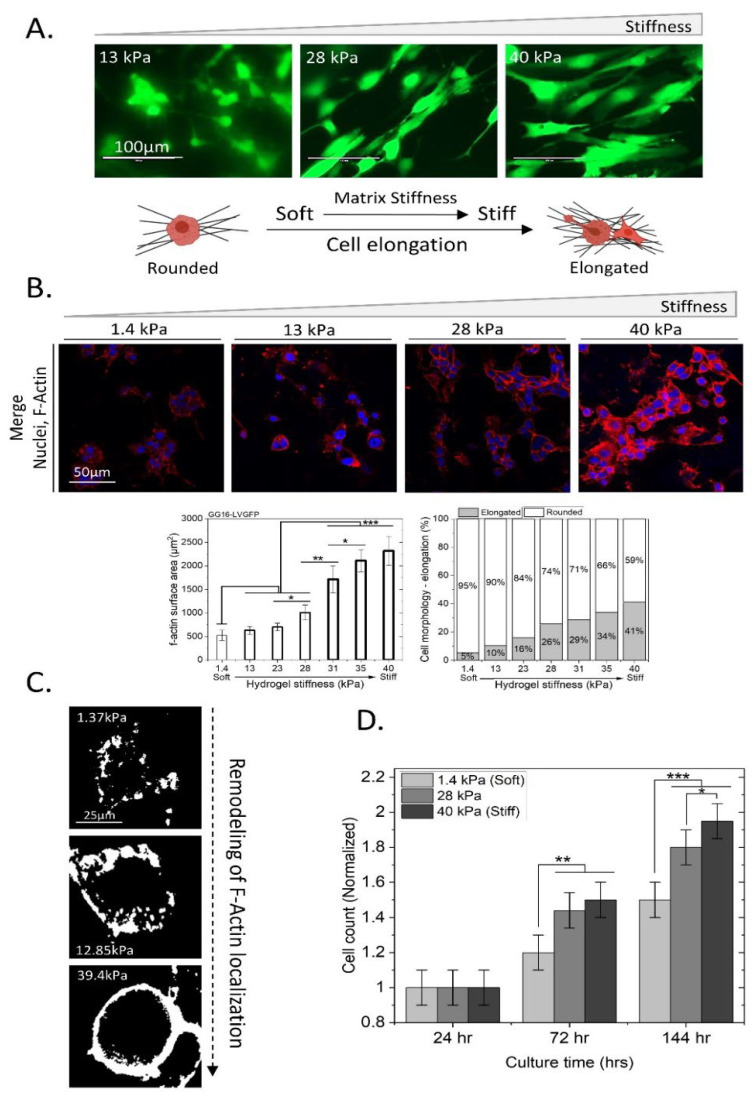
Adaptation of GG16-LVGFP cells to an increasingly stiff matrix involves changes in cell morphology, F-Actin expression, and proliferation. (**A**) GG16-LVGFP were grown on hydrogels at the indicated stiffness, and changes in cell morphology were detected using laser scanning confocal microscopy (40×). (**B**) Microscopic analyses of GG16 cells cultured for 6 days on hydrogels with increasing stiffness. Nuclei stained with Dapi^TM^ (Blue), and F-Actin stained with Alexa fluor^TM^ 546-Phalloidin (Red). F-Actin expression and cell elongation were quantified. (**C**) Magnified image of F-Actin remodeling in GG16 cells showing plasma membrane localization at highest stiffness. (**D**) GG16-LVGFP cell proliferation was determined at increasing hydrogel stiffnesses, showing elevated proliferation rates with increasing matrix stiffnesses. * *p* ≤ 0.05; ** *p* ≤ 0.01; *** *p* ≤ 0.001.

**Figure 3 ijms-23-06520-f003:**
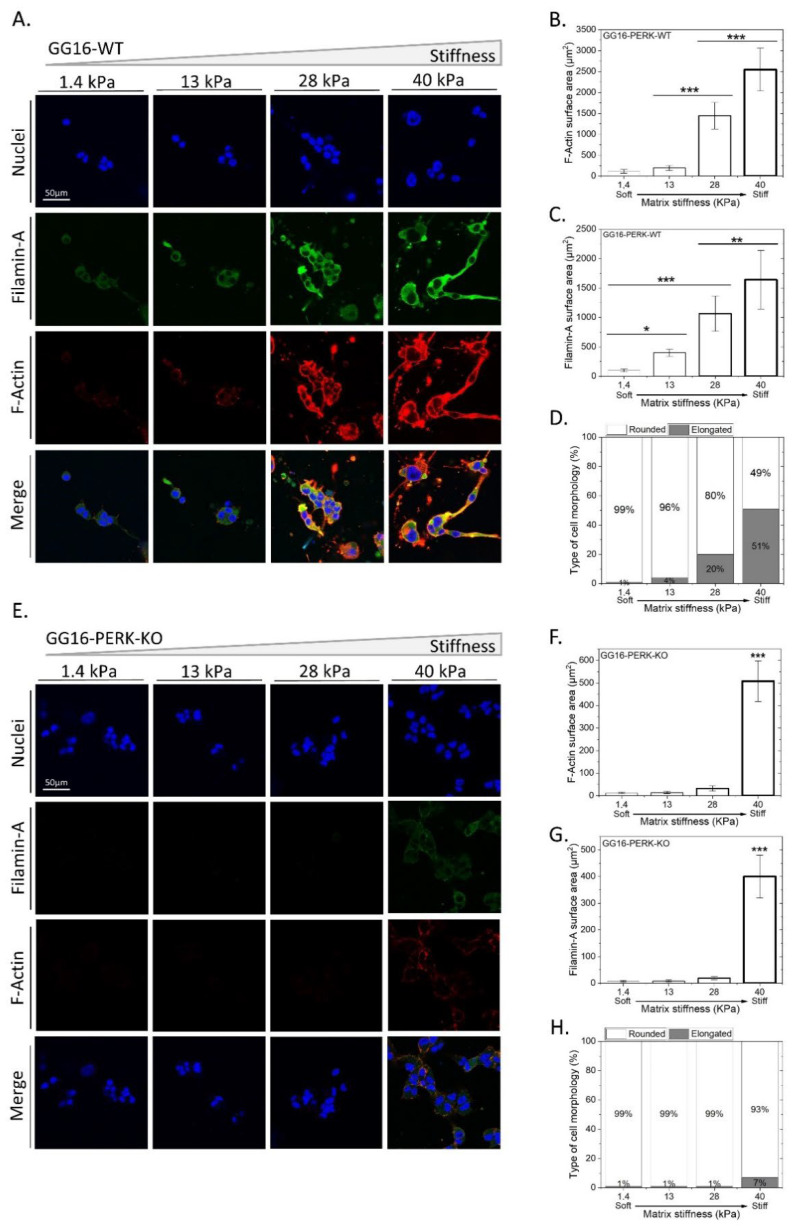
PERK-deficient GG16 cells are impaired in cellular adaptation to increasing stiffness which is linked with aberrant FLNA expression. (**A**,**E**) GG16-WT and PERK-KO cells were cultured for 6 days on hydrogels with different stiffnesses and stained with Dapi^TM^ (Blue), Alexa fluor^TM^ 546-Phalloidin (Red), and FLNA—Alexa fluor^TM^ 488 (Green). Cell morphology, F-Actin, and FLNA expression was quantified and is depicted in (**B**–**D**) and (**F**–**H**) for GG16-WT and PERK-KO cells, respectively. Cell morphology (from round to elongated), F-Actin, and FLNA expression altered gradually in a stiffness-dependent manner in GG16-WT cells, which was not seen in PERK-deficient cells. * *p* ≤ 0.05; ** *p* ≤ 0.01; *** *p* ≤ 0.001.

**Figure 4 ijms-23-06520-f004:**
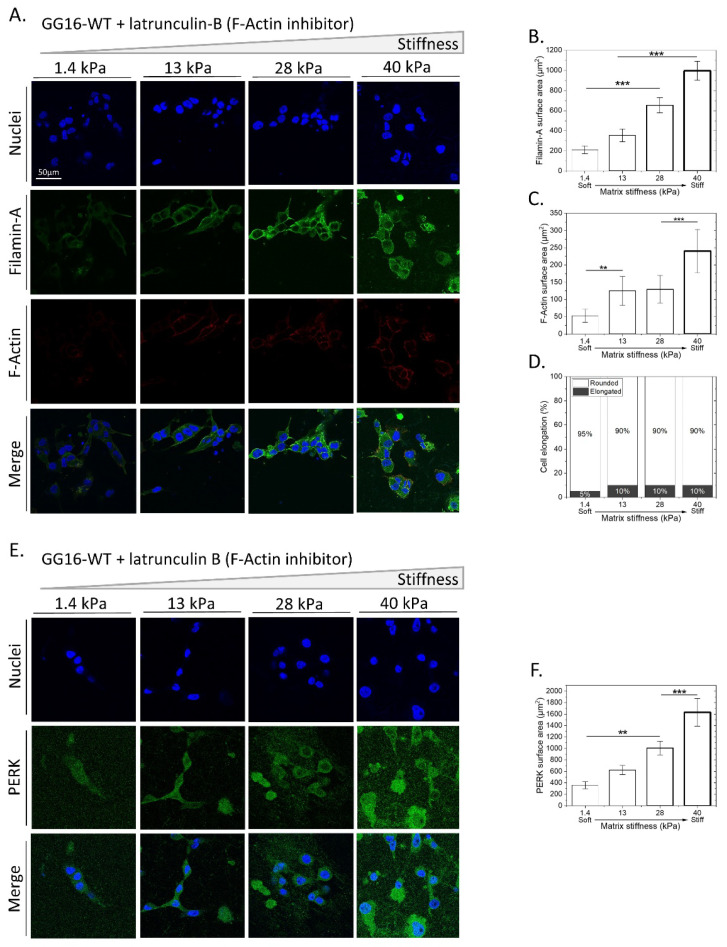
Inhibition of F-Actin polymerization mimics phenotype of PERK-deficient cells by impairing cellular adaption to matrix stiffness. (**A**,**E**) GG16-WT cells treated with latrunculin B were cultured for 6 days on hydrogels of varying stiffness. Cells were stained for F-Actin, FLNA, and PERK together with the nuclei and imaged with confocal microscopy. Specific fluorescence was quantified for FLNA (**B**), F-Actin (**C**), cell elongation (**D**), and PERK expression (**F**). ** *p* ≤ 0.01; *** *p* ≤ 0.001.

**Figure 5 ijms-23-06520-f005:**
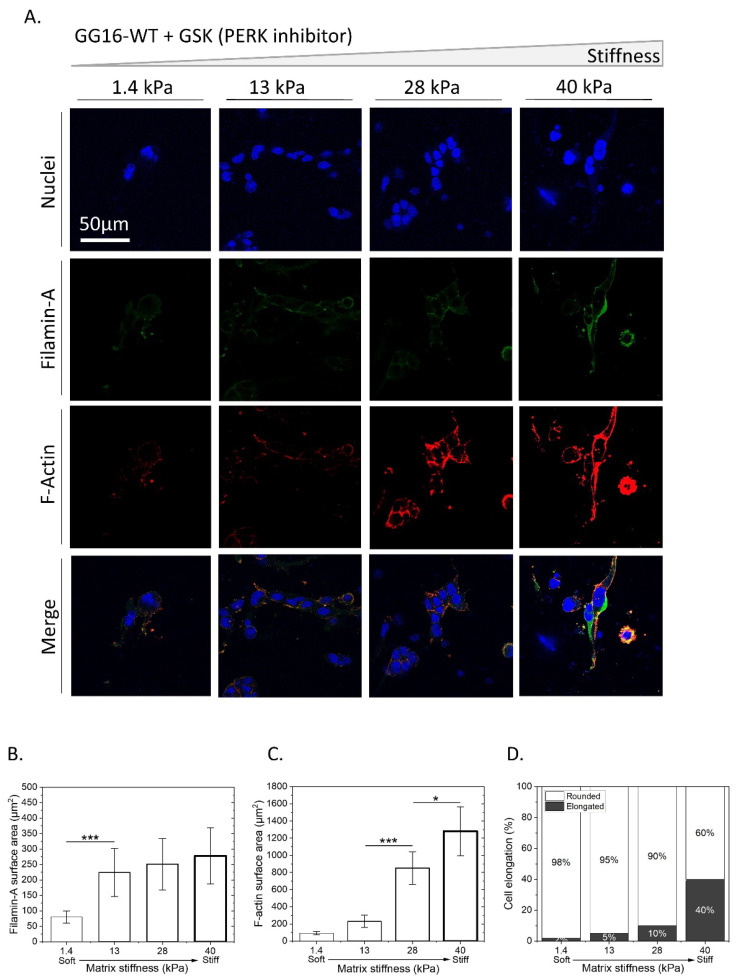
Inhibition of PERK kinase activity partially impairs stiffness-dependent cellular adaptation. GG16-WT cells were treated with GSK414 for 6 days while growing on hydrogels with increasing stiffness. (**A**) Cells were stained for FLNA and F-Actin and imaged with confocal microscopy. (**B**–**D**) Quantified FLNA, F-Actin expression, and cell elongation. * *p* ≤ 0.05; *** *p* ≤ 0.001.

**Figure 6 ijms-23-06520-f006:**
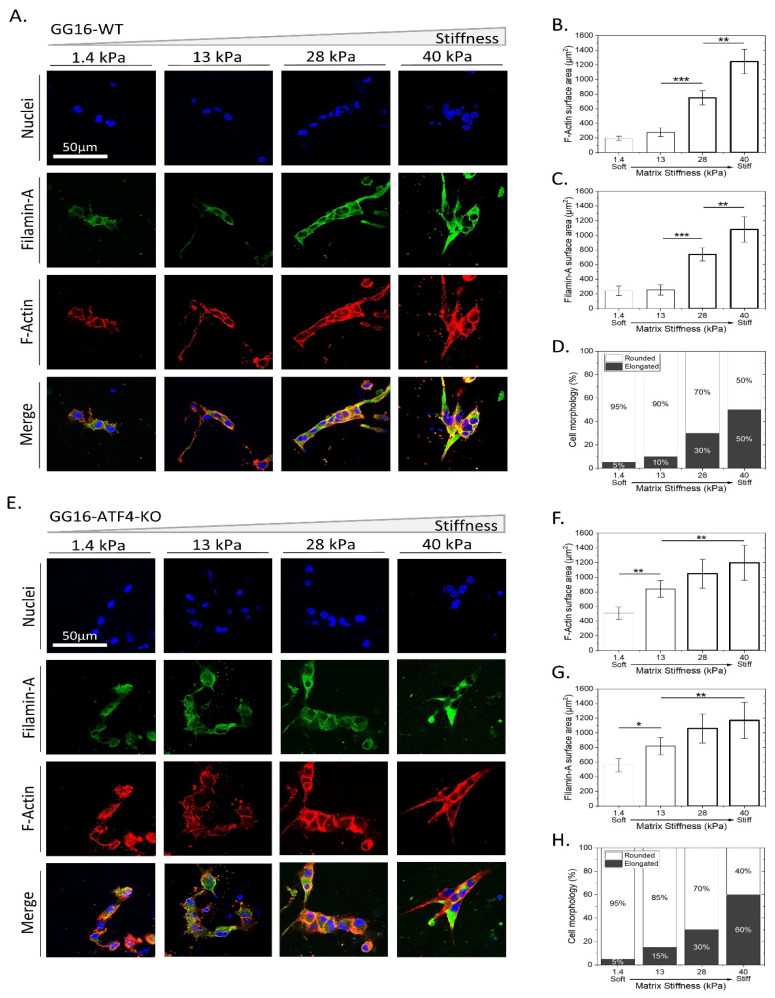
ATF4 is not involved in cellular adaption to matrix stiffness. GG16-WT and GG16-ATF4-KO cells were cultured on hydrogels with increasing stiffness. (**A**,**E**) Cells were stained for FLNA, F-Actin, and nuclei and imaged with confocal microscopy. Quantified expression of FLNA and F-Actin (**B**,**C**,**F**,**G**) and cell elongation (**D**,**H**). * *p* ≤ 0.05; ** *p* ≤ 0.01; *** *p* ≤ 0.001.

**Figure 7 ijms-23-06520-f007:**
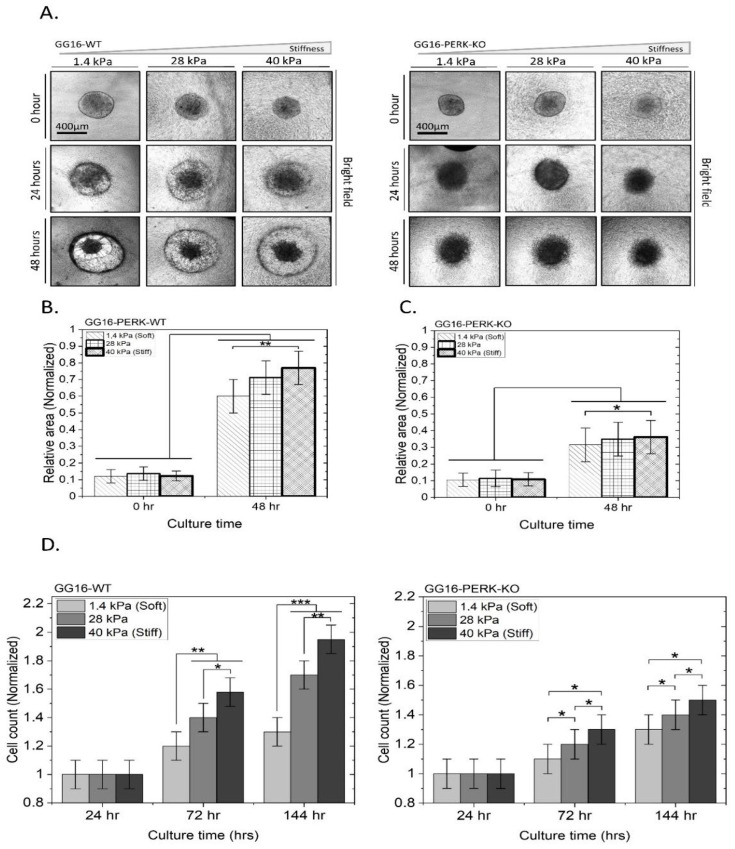
PERK deficiency reduces stiffness-dependent cell motility and proliferation. GG16-WT and PERK-KO spheroids were seeded on three hydrogels with increasing stiffness, as indicated (**A**). Cell migration was monitored during 48 h using brightfield microscopy. The surface area of the expanded spheroids was calculated and normalized to the surface area at 0 h (**B**,**C**). (**D**) Cell proliferation of GG16-WT and PERK-KO cells was determined after 6 days of growth on indicated stiff matrixes with Cell Tracker™ Green CMFDA Dye. Cell numbers were normalized to the 24 h point. * *p* ≤ 0.05; ** *p* ≤ 0.01; *** *p* ≤ 0.001.

**Figure 8 ijms-23-06520-f008:**
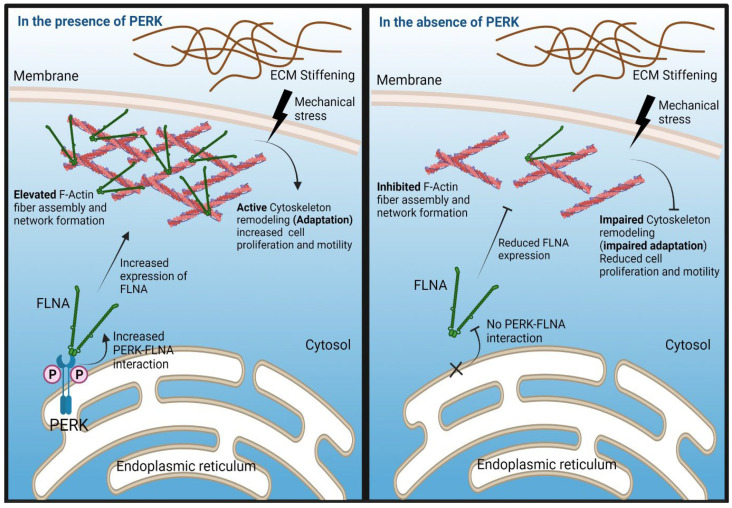
Model depicting the role of PERK in mediating cellular adaptation of GSC to increasing matrix stiffness. In PERK-proficient cells, increasing matrix stiffness results in increased PERK and FLNA expression. PERK-FLNA interactions are required for F-Actin remodeling which is essential for cellular adaptation to stiffness. Phosphorylation of PERK likely facilitates FLNA interaction. This is accompanied by a change in cell morphology from round to an elongated phenotype and increased cell proliferation and migration. In PERK-deficient cells, this mechanism regulating stiffness-dependent F-Actin remodeling is disrupted, resulting in lack of morphological change and no stimulation of proliferation and migration.

## Data Availability

Not applicable.

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
