# Peer review of "The Unfolded Protein Response Sensor PERK Mediates Stiffness-Dependent Adaptation in Glioblastoma Cells"

_ijms, 2022, doi:10.3390/ijms23126520_

Round 1
Reviewer 1 Report
Khoonkari and colleagues presented a research article aimed at investigating the involvement of PERK in the stiffness of the extracellular matrix in glioblastoma models. For this purpose, the authors used human blood plasma (HBP)/alginate hydrogels with different grades of stiffness and GBM cells to evaluate the involvement of PERK and other proteins in this process. Overall, the experimental design was well-conceived. However, there are some issues that the authors have to address before publication:
1) The authors have to shorten the Introduction section;
2) The authors have to include the statistical significance for the data obtained in the different experiments. This comment is mandatory and is valid for all the data shown in Figure 2D to Figure 7D;
3) It is not clear how the authors evaluated the modulation of PERK levels after cell transduction. Did the authors evaluate the mRNA expression levels and protein levels of PERK through RT-qPCR and Western blot, respectively? Please discuss these aspects;
4) Please confirm that 500 ml (and not uL) of human blood was obtained by healthy donors. Did the authors gain the approval of an ethics committee?
5) In the Discussion section, the authors have to emphasize the translational relevance of their findings. Could the evaluation of ECM stiffness be used to improve the diagnosis, prognosis or therapy of GBM? Please argue these aspects;
6) In Chapter 4.6. please provide the sequences or catalog numbers of the vectors used;
7) Consider to shorten Chapter 4.7;
8) Did the authors perform any statistical analyses? The authors should include a new chapter in the Methods section “Statistical Analyses”.
Author Response
Response to the reviewer’s comments
Manuscript title: The Unfolded Protein Response Sensor PERK Mediates Stiffness-dependent Adaptation in Glioblastoma Cells.
We thank the reviewer for their positive assessment of our paper and the insightful comments. Please find below our point-by-point response to the comments.
Reviewer 1:
- The authors have to shorten the Introduction section;
Authors response:
Thanks for the suggestion, however, we would like to keep the introduction as is. The research presented covers various aspects of cancer biology particularly GBM, cancer stem cells, TME/ ECM, mechanical stress/ stiffness and UPR that all need to be introduced and require sufficient explanation in the context of our research questions. We attempted to be brief and we believe the background provided is required to make it understandable for interested readers.
- The authors have to include the statistical significance for the data obtained in the different experiments. This comment is mandatory and is valid for all the data shown in Figure 2D to Figure 7D;
Authors response:
Thanks for the suggestion. We have adapted the graphs in Figure 2D and 7D and replaced these with bar graphs in order to indicate statistical significance of stiffness dependent changes in proliferation.
For the figure panels depicting the average percentages of elongated cells vs rounded cells at different hydrogel stiffness the data represent ratios between these phenotypes and statistics could not be indicated.
- It is not clear how the authors evaluated the modulation of PERK levels after cell transduction. Did the authors evaluate the mRNA expression levels and protein levels of PERK through RTqPCR and Western blot, respectively? Please discuss these aspects;
Authors response:
As indicated in methods section GG16 PERK-KO cells have been generated earlier using crispr/cas editing (Peñaranda-Fajardo, N.M., et al. Cell Death Dis 10, 690 (2019). https://doi.org/10.1038/s41419-019-1934-1, ref 33). Effective PERK-KO cells were selected based on absence of protein expression by western blotting. In addition, absence of PERK can also be appreciated in supplementary Figure S3, showing absence of PERK staining by immunofluorescent microscopy.
- Please confirm that 500 ml (and not uL) of human blood was obtained by healthy donors. Did the authors gain the approval of an ethics committee?
Authors response:
Thanks very much for the opportunity to describe this better. Indeed 500 ml of human blood plasma was obtained from 5 healthy (voluntary) donors, with the approval of ethics committee of the university medical center Groningen (UMCG) that is now indicated in methods 4.6. Also the following text has been added for clarification (page 22/line 583-586):
‘’Institutional Review Board Statement: The study was conducted in accordance with the Declaration of University Medical Center Groningen (UMCG), and approved by the Institutional Re-view Board (or Ethics Committee) of University Medical Center Groningen (UMCG) (protocol code: METc2013.135 and date of approval: 23.05.2013) for studies involving humans’’.
- In the Discussion section, the authors have to emphasize the translational relevance of their findings. Could the evaluation of ECM stiffness be used to improve the diagnosis, prognosis or therapy of GBM? Please argue these aspects;
Authors response:
Thanks for the suggestion. Our research was not directly aimed at examining ECM stiffness in relation to GBM aggressiveness, but was primarily focused on examining the role of PERK in cellular adaptation to mechanical stress/ stiffness. To make a link with clinical relevance we have added the following text to the Discussion (page 18/line 383-385):
‘The therapeutic relevance and implications of our findings remain to be further investigated. However, it is clear that therapeutic modulation of PERK not only targets the UPR, but will also affect cancer cell stemness and plasticity, which also includes the adaptation to mechanical stress.’
- In Chapter 4.6. please provide the sequences or catalog numbers of the vectors used;
Authors response:
The source of the pRRL-SFFV-IRES-EGFP vector has been indicated and an additional reference was included (Wierenga et al 2014, https://doi.org/10.1371/journal.pone.0093494. (page 19/line 439-440)
- Consider to shorten Chapter 4.7;
Authors response:
Chapter 4.7 has been shortened.
- Did the authors perform any statistical analyses? The authors should include a new chapter in the Methods section “Statistical Analyses”;
Authors response:
A new chapter 4.10 (page 21) has been added to describe the statistical analyses. To avoid repetitive statistical statements were removed in methods section.

Reviewer 2 Report
Khoonkari et al. submitted a fascinating manuscript describing the role of PERK protein in stiffness-dependent adaptation in the GBM model. The manuscript is generally well-written, and the presented results support the conclusions. The authors found that PERK protein, a known UPR sensor, also mediates the GBM cells' response to ECM stiffening. The authors found that downstream PERK-regulated ATF4 protein was not activated. ATF4 expression is mediated via IRE1a-dependent PERK activation. Was IRE1a activated in PERK-induced cells? What about CHOP protein? What is the role of UPR in this process?
Minor issues:
- Please provide for all reagents the information about the company name, town, and country. In the case of the US, also state.
Author Response
Response to the reviewer’s comments
Manuscript title: The Unfolded Protein Response Sensor PERK Mediates Stiffness-dependent Adaptation in Glioblastoma Cells.
We thank the reviewer for their positive assessment of our paper and the insightful comments. Please find below our point-by-point response to the comments.
Reviewer 2:
- Was IRE1a activated in PERK-induced cells? What about CHOP protein? What is the role of UPR in this process?
Authors response:
We focused on studying role of PERK in stiffness adaptation. It would indeed be interesting to examine if and how IRE1a is related to PERK activity in this process. Particularly since IRE1 also has been demonstrated to interact with and regulate FLNA as we mention in discussion. However, this is out of scope of our current study and will be part of further investigations. We found a minor role for PERK kinase activation in stiffness adaptation, whereas ATF4 was not involved as determined by use of GG16-ATF4-KO cells. Although we did not look at CHOP, which has been linked with cell death activation, our data indicate no involvement of the classical UPR and points rather to involvement of PERK scaffold function (PERK-FLNA protein interactions) in the adaptive response.
- Please provide for all reagents the information about the company name, town, and country. In the case of the US, also state.
Authors response:
The text is modified based on the reviewer suggestion with adding the company name, town and country for all reagents.
Pages 18-19, methods and materials are all corrected by adding the relative information for the reagents.

Reviewer 3 Report
The findings appear to be interesting and technically well performed. Specific points that the authors need to address are as follows:
1. Most of the experiments have been done in GBM stem cells. A limited in vivo study will greatly increase the impact of the findings.
2. The mechanisms that lead to increase in proliferation and F-Actin polymerization upon increasing stiffness should be analyzed.
3. Whether deletion of PERK by si-RNA can also impair stiffness-dependent cellular adaptation should be analyzed.
4. Whether PERK can also regulate migration and invasion capability of GBM cells can be investigated.
5. Typographical errors were found throughout the manuscript and should be corrected by using professional editing service.
6. The authors should provide their own justification and relevance of the study. This will help the readers to understand the importance of the paper.
Author Response
Response to the reviewer’s comments
Manuscript title: The Unfolded Protein Response Sensor PERK Mediates Stiffness-dependent Adaptation in Glioblastoma Cells.
We thank the reviewer for their positive assessment of our paper and the insightful comments. Please find below our point-by-point response to the comments.
Reviewer 3:
- Most of the experiments have been done in GBM stem cells. A limited in vivo study will greatly increase the impact of the findings.
Authors response:
Indeed, performing in vivo studies would further shed light on the role of PERK in GBM formation and progression and in the context of ECM related stiffening. However, setting up and performing animal studies is time consuming and costly and is planned as follow up research. Moreover, to study brain ECM stiffening in relation to GBM adaptation and the role of PERK herein requires considerable investments in developing and validating relevant methodology. Thus, although this will be very relevant, currently this is out of scope of the current work and remains to be examined in follow up research.
- The mechanisms that lead to increase in proliferation and F-Actin polymerization upon increasing stiffness should be analyzed.
Authors response:
The finding that increased matrix stiffening enhances proliferation is indeed interesting. In literature increasing matrix stiffness has been linked with enhanced cancer cell proliferation, as reviewed by Kalli et. al. 2018. https://doi.org/10.3389/fonc.2018.00055.
Studies reported increased GBM cell proliferation upon matrix stiffening using 3D in vitro hydrogel models such as for example Wang et al 2020, https://doi.org/10.1089/ten.tea.2020.0110.
It is known that matrix stiffness can activate various intracellular signaling pathways to regulate cellular behavior, including focal adhesion complexes and downstream mechanisms. Currently we are looking more closely on how different signaling pathways could be affected upon altered stiffness in cells with/out PERK and is the topic of future work.
- Whether deletion of PERK by si-RNA can also impair stiffness-dependent cellular adaptation should be analyzed.
Authors response:
Silencing of PERK and other relevant genes by specific siRNA is indeed an attractive approach. We have tested this approach in our GSC models, however, encountered some difficulties in obtaining sufficient transfection efficiencies and levels of downregulation. Another complication is that siRNA mediated downregulation last for several days, whereas our experiment can take up to 7 days. For now we have used crispr/cas editing and pharmacological inhibitors to modulate PERK.
- Whether PERK can also regulate migration and invasion capability of GBM cells can be investigated.
Authors response:
This is indeed an interesting point and which we actually examined in this work. For this purpose we made spheroids from GG16-WT and GG16-PERK-KO cells, seeded on top of hydrogels with different stiffnesses and migration was monitored over 48 hours. This experiment showed that PERK regulates GSCs motility, as the invasive front of the cells with PERK was more than 2 folds higher compare to cells without PERK (see Figure 7 and discussion). The molecular basis for these observation is further examined although reduced F-Actin remodeling in PERK-KO cells can be assumed to have a severe impact on cell motility.
- Typographical errors were found throughout the manuscript and should be corrected by using professional editing service.
Authors response:
Our apologies for this. The paper was checked again and identified typos were corrected.
- The authors should provide their own justification and relevance of the study. This will help the readers to understand the importance of the paper.
Authors response:
Thanks for the suggestion. To make a link with clinical relevance we have added the following text to conclude the Discussion (page 18/line 383-385):
‘The therapeutic relevance and implications of our findings remain to be further investigated. However, it is clear that therapeutic modulation of PERK not only targets the UPR, but will also affect cancer cell stemness and plasticity, which also includes the adaptation to mechanical stress.’

Round 2
Reviewer 1 Report
The authors well addressed all my previous comments. The manuscript can be accepted for publication after the editorial check.